# A Framework for Cluster and Classifier Evaluation in the Absence of Reference Labels

**Robert J. Joyce, Edward Raff**
Booz Allen Hamilton
Univ. of Maryland, Baltimore County

**Charles Nicholas**
Univ. of Maryland, Baltimore County

## Abstract

In some problem spaces the high cost of obtaining ground truth labels necessitates use of lower quality reference datasets. It is difficult to benchmark model changes using these datasets, as evaluation results may be misleading or biased. We propose a supplement to using reference labels which we call an approximate ground truth refinement (AGTR). Using an AGTR we prove that bounds on the precision and recall of a clustering algorithm or multiclass classifier can be computed without reference labels. We introduce a litmus test that uses an AGTR to identify inaccurate evaluation results produced from reference datasets of dubious quality. Creating an AGTR requires domain knowledge, and malware family classification is a task with robust domain knowledge approaches that support the construction of an AGTR. We demonstrate our AGTR evaluation framework by applying it to a popular malware labeling tool to diagnose over-fitting in prior testing and evaluate changes that could not be meaningfully quantified in their impact under previous data.

## 1 Introduction

The capabilities of a new clustering algorithm or classifier must be assessed both during and after development. Various metrics are used to evaluate these capabilities so that an end user can make an informed choice about which model is most suitable for their needs or fine-tune the parameters of a chosen approach. A *reference dataset* is required for computing these metrics. Each data point in a reference dataset has a *reference label*, which is the correct label that a classifier is expected to predict. When evaluating a clustering algorithm, we refer to a dataset grouped by reference label as a *reference clustering*. Not all reference datasets are equally fit for performing evaluation. As we will later discuss in Appendix C, reference datasets can be too small, lack diversity, or have an imbalanced class distribution. In some fields, it is not feasible to obtain ground truth reference labels for a large dataset, so small reference datasets or datasets with lower quality reference labels are used in their stead. Using reference datasets that have these deficiencies may produce inaccurate or misleading evaluation results [1]. However, when a field has no satisfactory reference datasets, how can one trust evaluation results? This problem makes it difficult to determine which (if any) model is most suitable for a task, or if progress is being made while developing a model.

Our work provides an improvement to these undesirable circumstances by introducing a provable framework for quantifying performance in the absence of reference labels. Our approach can be used to check for over-fitting in benchmark design, sanity-check labeling procedures, and compare the performance of intrinsically similar models. We will finish this introduction with the specific terminology and definitions that our framework will operate in. These are used in § 2 to develop an *approximate ground truth refinement* (AGTR). The AGTR of a dataset is a (incomplete) sub-graph of the ground truth reference labels where links in the AGTR indicate a positive relationship, but the absence of a link does not imply the absence of a relationship. In subsection 2.4 we will describe the general utility of AGTRs across any application, and in § 3 we will refine the discussion to malware

Submitted to the 35th Conference on Neural Information Processing Systems (NeurIPS 2021) Track on Datasets and Benchmarks. Do not distribute.

specific applications and needs. In §4 we will evaluate the seminal AVClass tool and the impact of various changes, which could not be previously elucidated due to a lack of precise labels. In doing so we will demonstrate evidence that the original benchmark may have over-fit to noisy reference labels, and quantify the impact of design choices that could not previously be quantified meaningfully with public data. We will conclude and discuss limitations in §5.

## 1.1 Metric Terminology

Before we can discuss the AGTR evaluation framework we must first introduce the terminology used in this paper. Let $M$ be a dataset consisting of $m$ unique data points. Let $C = \{C_i\}_{1 \le i \le c}$ and $D = \{D_j\}_{1 \le j \le d}$ each partition $M$, where $C$ is the predicted clustering of the dataset and $D$ is the reference clustering. Let $f : \{1...c\} \mapsto \{1...d\}$ and $g : \{1...d\} \mapsto \{1...c\}$ be functions mapping the predicted labels to the reference labels and vice versa. The label translation functions $f$ and $g$ are defined differently for clustering and classification problems. When evaluating clustering algorithms, no labels exist which can map between clusters in $C$ and $D$. Instead, $f$ and $g$ are defined as $f(i) = \underset{j}{\arg\max} |C_i \cap D_j|$ and $g(j) = \underset{i}{\arg\max} |C_i \cap D_j|$ [2].

These function definitions map each predicted cluster to the reference cluster for which there is maximal overlap and vice versa. When evaluating a classifier, the set of labels used by the classifier is typically equivalent to the set of labels used by the reference dataset. In these cases, $c = d$, and $f$ and $g$ are defined as the identity function [2]: $\forall i, 1 \le i \le c, f(i) = i$ and $g(i) = i$. When the labels used by the classifier do not map directly those used in the reference dataset, either a custom mapping or the function definitions for mapping clusters are used.

## 1.2 Computing Precision, Recall, and Accuracy

In this paper we discuss three metrics used for evaluating clustering algorithms and multiclass classifiers: precision, recall, and accuracy. Historically, precision and recall have been used for evaluating the performance of information retrieval systems [3]. Bayer *et al.* [4] introduced alternate definitions of precision and recall as cluster validity indexes. Li *et al.* [2] broadened these definitions to allow for evaluation of multiclass classifiers. Additionally, Li *et al.* show that the accuracy of a classifier can be computed as a special case of precision and recall. We now discuss the precision, recall, and accuracy metrics and how they are computed.

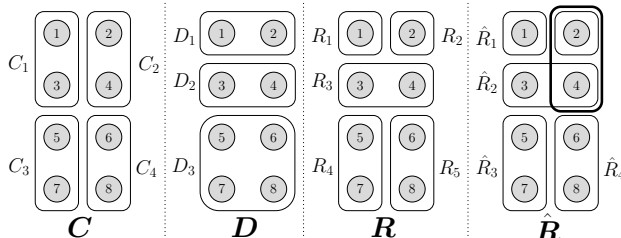

Figure 1: Four partitions of a hypothetical dataset. The predicted clusters ("$C$") would ideally be evaluated using ground truth ("$D$"). A GTR ("$R$") informs a subset of the data point relationships in $D$ (e.g., 5 and 7 ($R_4$) must belong to the same reference cluster, but without $D$ it is unknown whether the members of $R_4$ and $R_5$ share a reference cluster). An AGTR ("$\hat{R}$") is a GTR with $\epsilon$ errors. $\hat{R}_2$ incorrectly groups data point 2 with 3 and 4, so it has $\epsilon = 1$ errors. If data point 2 is removed from $\hat{R}_2$, $\hat{R}$ becomes a GTR.

**Definition 1.** $Precision(C, D) = \frac{1}{m} \sum_{i=1}^{c} |C_i \cap D_{f(i)}|$

**Definition 2.** $Recall(C, D) = \frac{1}{m} \sum_{j=1}^{d} |C_{g(j)} \cap D_j|$

When used as a cluster validity index, *Precision* measures how well a clustering separates data points belonging to different reference clusters. Precision penalizes the presence of impure clusters, *i.e.*, clusters containing data points belonging to separate reference clusters [2]. A high precision (near one) indicates that few clusters are impure while a low precision (near zero) indicates that many clusters are impure. Precision tends to become inflated as the number of predicted clusters increases. The precision of a clustering is one if every data point is assigned to its own cluster because no cluster is impure [4]. In Figure 1, *Precision(C, D)* = 0.75.

*Recall* measures how well a clustering groups data points belonging to the same reference cluster. Recall penalizes instances in which data points belonging to the same reference cluster do not appear in the same predicted cluster [2]. Contrary to precision, recall may become inflated as the number of predicted clusters decreases. The recall of a clustering is one if all data points are grouped in a single cluster because no data points with the same reference label belong to different predicted clusters [4]. In Figure 1, *Recall(C, D)* = 0.5.

**Definition 3.** *If $f$ and $g$ are the identity function, Accuracy(C, D) = Precision(C, D) = Recall(C, D)*

*Accuracy* measures how frequently the predicted label matches the reference label. By Definition 3 accuracy, precision, and recall are all equivalent when $f$ and $g$ are the identity functions [2]. Accuracy cannot be computed if there is not a one-to-one mapping between the predicted clusters and the reference clusters, such as in Figure 1.

# 2 Approximate Ground Truth Refinements

In this section we introduce the concept of a ground truth refinement (GTR) and show that a GTR can be used to find provable bounds on precision, recall, and accuracy. In practice, when constructing a GTR from a dataset we assume that a small number of errors occur. We call an imperfect GTR an approximate ground truth refinement (AGTR). We show that bounds on these evaluation metrics can still be proven using an AGTR if errors in the AGTR construction process are properly accounted for. Finally, we propose a framework that uses an AGTR to evaluate clustering algorithms and multiclass classifiers when satisfactory reference data is unavailable. All proofs are located in Appendix D.

## 2.1 Set Partition Refinements

A key element of this work is the concept of a set partition refinement. Suppose two partitions $R$ and $S$ of the same set $M$. $R$ is a *refinement* of $S$ if each set within $R$ is a subset of some set in $S$ [5].

**Definition 4.** *If $\forall R_k \in R, \exists S_j \in S$ s.t. $R_k \subseteq S_j$ then $R$ is a set partition refinement of $S$.*

Set partition refinements can also be considered from an alternate perspective. If $R$ is a refinement of $S$, then $S$ can be constructed by iteratively merging sets within $R$. Specifically, each set $S_j \in S$ is equivalent to the union of some unique set of sets within $R$.

**Property 1.** $\forall S_j \in S, \exists! Q_j = \{Q_{j\ell}\}_{1 \leq \ell \leq q_j}$ s.t. $S_j = \bigcup\limits_{\ell=1}^{q_j} Q_{j\ell}$ and $\forall Q_{j\ell} \in Q_j, Q_{j\ell} \in R$

In Section 2.2, we use Definition 4 and Property 1 to prove properties of ground truth refinements, which are a type of set partition refinement.

## 2.2 Ground Truth Refinements

A *ground truth refinement* (GTR) of a dataset is a clustering where all data points in a cluster are members of the same ground truth reference cluster. Importantly, the opposite is not necessarily true, as data points in the same reference cluster can belong to different clusters in the GTR.

**Definition 5.** *If $D$ is a ground truth reference clustering and $R$ is a refinement of $D$, then $R$ is a ground truth refinement.*

Recall that for a dataset $M$, $C = \{C_i\}_{1 \leq i \leq c}$ is the predicted clustering and $D = \{D_j\}_{1 \leq j \leq d}$ is the reference clustering. Let $D$ have ground truth confidence and let $R = \{R_k\}_{1 \leq k \leq r}$ be a GTR of $D$. Since $R$ partitions $M$, it is possible to compute the precision and recall of $C$ with respect to $R$ rather than $D$. An important trait of a GTR is that it does not require reference labels. Since $R$ is unlabeled, we map each predicted cluster in $C$ to the cluster in the $R$ for which there is maximal overlap and vice versa. Let the functions for mapping between the predicted clusters and the GTR $f' : \{1...c\} \mapsto \{1...r\}$ and $g' : \{1...r\} \mapsto \{1...c\}$ be defined as $f'(i) = \operatorname*{argmax}_k |C_i \cap R_k|$ and $g'(k) = \operatorname*{argmax}_i |C_i \cap R_k|$.

Using Definition 4 we prove that the precision of a clustering algorithm or multiclass classifier computed using the ground truth reference clustering is bounded below by its precision computed

using a GTR. Similarly, using Property 1 we prove that recall computed using a GTR is always an upper bound on recall computed using the ground truth reference clustering. Because accuracy, precision, and recall are all equivalent in a special case, we prove that recall computed using a GTR is also always an upper bound on the accuracy of a classifier. These bounds provide the foundation for evaluating clustering algorithms and multiclass classifiers using an AGTR.

**Theorem 1.** *Precision(C, R) $\leq$ Precision(C, D)*      **Theorem 2.** *Recall(C, R) $\geq$ Recall(C, D)*

**Corollary 2.1.** *Recall(C, R) $\geq$ Accuracy(C, D)*

## 2.3 Approximate Ground Truth Refinements

Unfortunately, it is impossible to confirm whether or not a clustering is a GTR without knowing the ground truth reference clustering. Because we intend for GTRs to be used when satisfactory reference datasets are not available, this is problematic. When attempting to construct a GTR, we assume that the resulting clustering is very similar to a GTR but has a small number of data points $\epsilon$ which violate the properties of a refinement. We call such a clustering an *approximate ground truth refinement* (AGTR).

**Definition 6.** *If $R$ is a ground truth refinement and $\hat{R}$ can be made equivalent to $R$ by correcting the cluster membership of $\epsilon$ data points, then $\hat{R}$ is an approximate ground truth refinement.*

Suppose an AGTR $\hat{R}$ with $\epsilon$ erroneous data points. Even without knowing which data points must be corrected to transform $\hat{R}$ into a GTR, we can again derive bounds on *Precision(C, D)* (abbreviated as "Prec" when needed) as well as upper bounds on *Recall(C, D)* and *Accuracy(C, D)* using $\hat{R}$ and $\epsilon$. To do this, we first show that the precision and recall change in predictable ways when a reference clustering is modified. Let $S$ be an arbitrary partition of a dataset $M$ and let $\hat{S}$ be identical to $S$ but with a single data point belonging to a different cluster. When the precision and recall of $C$ are measured with respect to $S$ and $\hat{S}$, the metrics share the following relationship:

**Theorem 3.** $|Prec(C, S) - Prec(C, \hat{S})| \leq \frac{1}{m}$      **Theorem 4.** $|Recall(C, S) - Recall(C, \hat{S})| \leq \frac{1}{m}$

Theorems 3 and 4 show that precision and recall can vary by up to $\pm\frac{1}{m}$ when the cluster membership of a single data point in the reference clustering is changed. Therefore, if $\epsilon$ cluster labels in $\hat{R}$ are erroneous, the difference between *Precision(C, R)* and *Precision(C, $\hat{R}$)* as well as between *Recall(C, R)* and *Recall(C, $\hat{R}$)* is at most $\pm\frac{\epsilon}{m}$

**Corollary 3.1.** $|Prec(C, R) - Prec(C, \hat{R})| \leq \frac{\epsilon}{m}$   **Corollary 4.1.** $|Recall(C, R) - Recall(C, \hat{R})| \leq \frac{\epsilon}{m}$

Unfortunately, these relationships require knowledge of the exact value of $\epsilon$, which is impossible to determine without knowing the ground truth reference clustering. Again, because the purpose of an AGTR is to be used when an adequate reference clustering is unavailable, this presents a problem. The solution is to select some value $\hat{\epsilon}$ with the belief that $\hat{\epsilon} \geq \epsilon$. We show that if this belief is true the bounds on precision, recall, and accuracy are valid.

**Theorem 5.** *If $\hat{e} \geq e$ then Precision(C, $\hat{R}$) $- \frac{\hat{\epsilon}}{m} \leq$ Precision(C, D)*

**Theorem 6.** *If $\hat{e} \geq e$ then Recall(C, $\hat{R}$) $+ \frac{\hat{\epsilon}}{m} \geq$ Recall(C, D)*

**Corollary 6.1.** *If $\hat{e} \geq e$ then Recall(C, $\hat{R}$) $+ \frac{\hat{\epsilon}}{m} \geq$ Accuracy(C, D)*

They allow the bounds on the precision, recall, and accuracy of a clustering algorithm or a multiclass classifier to be computed without reference labels.

## 2.4 Estimating Errors in an AGTR

We emphasize that the evaluation metric bounds from Theorem 5, Theorem 6, and Corollary 6.1 only hold if errors during AGTR construction are accounted for properly, *i.e.* $\hat{\epsilon} \geq \epsilon$. Selecting a satisfactory value of $\hat{\epsilon}$ for an AGTR is a matter of epistemic uncertainty and is an issue for future work. Determining the approximate error rate of a process used to construct an AGTR will likely require some guesswork, as a quality reference dataset is presumably unavailable. Domain experts should model their uncertainty about the AGTR construction method's error rate and choose a value of $\hat{\epsilon}$ that they believe exceeds the number of errors with very high confidence. In subsection 3.1 we provide an example of how to evaluate the error rate of an AGTR in order to select a judicious value of $\hat{\epsilon}$.

## 2.5 Properties of an Ideal AGTR

Although we have proven that it is possible to compute bounds on precision, recall, and accuracy using an AGTR, we have not yet proposed any techniques for constructing an AGTR from a dataset. Constructing an AGTR requires applying domain knowledge from a problem space to group data points with a high likelihood of sharing a reference label. Because of the domain knowledge requirement, no single technique can be used for general AGTR construction. Instead, a method for constructing an AGTR is specific to one kind of classification or clustering problem.

Some AGTR construction techniques will produce more useful evaluation metric bounds than others. Suppose a GTR $R$ constructed by simply assigning every data point to its own singleton cluster. We know that this method will always form a GTR with no errors because each singleton cluster must be the subset of some cluster in the ground truth reference clustering. However, when we use this GTR to compute the precision and recall of the predicted clustering, we obtain $Precision(C, R) = \frac{c}{m}$ and $Recall(C, R) = 1$, where $c$ is the number of predicted clusters. This GTR will never be useful for evaluation because these metric bounds are uninformative.

We have found that the similarity in composition between an AGTR and the reference clustering strongly influences the tightness or looseness of evaluation metric bounds. Given a ground truth reference clustering $D$ and an AGTR $\hat{R}$, let $\delta$ be the minimum number of data points in $\hat{R}$ whose cluster membership must be changed in order to transform it into $D$. Using Theorem 3 and Theorem 4 we show that the difference between a metric bound (prior to accounting for $\hat{\epsilon}$) and the true value of that metric is no greater than $\frac{\delta}{m}$.

**Corollary 3.2.** $|Prec(C, D) - Prec(C, \hat{R})| \leq \frac{\delta}{m}$ **Corollary 4.2.** $|Recall(C, D) - Recall(C, \hat{R})| \leq \frac{\delta}{m}$

Because evaluation metric bounds can deviate by up to $\frac{\delta}{m}$ from the true metric values, AGTRs that have smaller values of $\frac{\delta}{m}$, *i.e.*, ones that are as similar to the ground truth reference clustering as possible, are preferred. This allows us to identify the following three overall properties that should be considered in designing an AGTR in order for it to produce meaningful evaluation results:

*Low false positive rate.* An AGTR construction technique should group data points from different ground truth reference clusters as infrequently as possible. An increased rate of these false positives must be accounted for with a larger value of $\hat{\epsilon}$ to ensure that $\hat{\epsilon} \geq \epsilon$. This is undesirable, since a larger value of $\hat{\epsilon}$ results in looser evaluation metric bounds.

*Acceptable false negative rate.* A method for constructing an AGTR should be effective at grouping together data points with the same ground truth reference label. An AGTR with too many ungrouped data points will have a large value of $\delta$, resulting in loose bounds.

*Scalable.* Datasets used for constructing an AGTR should be large enough to adequately represent the problem space. A technique for constructing an AGTR must have acceptable performance when applied to a large number of data points.

## 3 Evaluating Malware Classifiers Using an AGTR

A malware family is a collection of malicious files that are derived from a common source code. Developing malware family classifiers is a substantial research area in the field of malware analysis [6]. However, current reference datasets are inadequate for accurately evaluating malware family classifiers and can cause biased or inaccurate evaluation results [1]. A major factor contributing to this issue is that obtaining ground truth family labels for malware is extremely time consuming. Although accurately determining the family of a malware sample is difficult, methods for automatically grouping similar malware samples together with low rates of error have been developed. Therefore, we believe that the process of evaluating malware clustering algorithms and malware family classifiers can greatly benefit from the AGTR evaluation framework. In this section we discuss the methods that are used to obtain malware reference labels, the datasets that have historically been used for classifier evaluation, and the issues that cause uncertainty in evaluation results when such approaches are used. We propose a method for constructing an AGTR from a dataset of malware samples and apply the AGTR evaluation framework with the approaches described in Appendix B to a popular malware classifier. We hope that this section provides a template for utilizing the AGTR evaluation framework in other clustering and classification problem spaces.

### 3.1 Constructing a Malware Dataset AGTR

In this section we discuss a method for constructing an AGTR from a dataset of malware samples. Because our method is automatic and scalable, the resulting AGTR can be orders of magnitude larger than a ground truth malware reference dataset and can include modern malware samples. Our method for constructing an AGTR from a malware dataset is based on peHash, a metadata hash for files in the Portable Executable (PE) format. Files in the PE format are executable files that can run on the Windows operating system, such as .exe, .dll, and .sys files. peHash was designed for identifying polymorphic malware samples within the same family as well as nearly identical malware samples. The hash digest is computed using metadata from the PE file header, PE optional header, and each PE section header [7]. Two malware samples with identical values for all of the chosen metadata features have identical peHash digests. Due to the number of metadata features used in the hash and the large range of possible values that these features can have, the odds that two unrelated malware samples share a peHash digest is minuscule.

Our proposed method for constructing an AGTR from a Windows malware dataset requires computing the peHash digest of each malware sample. Then, all malware samples that share a peHash digest are assigned to the same cluster. If the peHash of a malware sample cannot be computed, such as due to malformed PE headers, it is assigned to a singleton cluster. Using a hash table to tabulate clusters allows an AGTR to be built very efficiently, requiring only $O(m)$ memory usage and $O(m)$ run time complexity, where $m$ is the number of malware samples in the dataset.

Wicherski [7] evaluated the false positive rate of peHash using 184,538 malware samples from the mwcollect Alliance dataset and 90,105 malware samples in a dataset provided by Arbor Networks. All malware samples were labeled using the ClamAV antivirus engine [8]. The peHash of each malware sample in both datasets was calculated, resulting in 10,937 clusters for the mwcollect Alliance dataset and 21,343 clusters for the Arbor Networks dataset. Of these clusters, 282 and 322 had conflicting antivirus labels respectively. However, manual analysis showed that none of the clusters with conflicting antivirus labels contained unrelated malware samples. The evaluation method Wicherski used does not rule out the possibility of false positives. However, it is evident that the false positive rate of peHash is extremely low. Based on Wicherski's evaluation and our own additional assessment, we suggest choosing an $\hat{\epsilon}$ of approximately one percent the total dataset size when using a peHash AGTR. We believe that this value should far exceed the true number of errors $\epsilon$ in the AGTR.

A major consideration in the selection of peHash as our proposed AGTR construction method is its prevalent industry use. peHash is widely regarded to have an extremely low false positive rate. Furthermore, due to the adversarial nature of the malware ecosystem, Wicherski [7] has already analyzed peHash's vulnerabilities, and its widespread usage in industry means practitioners are aware of the real-world occurrence of attacks against it. These factors allow us to be very confident in our assessment of peHash's error rate. It was for these reasons that we elected to use peHash rather than design a custom AGTR construction technique. Developing new methods for constructing AGTRs is a target of future work that may yield tighter evaluation metric bounds.

## 4 Applying the AGTR Evaluation Framework to AVClass

At this point we have established the AGTR evaluation framework, discussed how malware classifier evaluation can benefit from it, and introduced a method for constructing an AGTR from a dataset of Windows malware using peHash. We will now apply the AGTR evaluation framework to the malware labeling tool AVclass [9]. When provided an antivirus scan report for a malware sample, AVClass attempts to aggregate the many antivirus signatures in the report into a single family label. AVClass is open source, simple to use, and does not require the malware sample to obtain a label, making it a popular choice as a malware classifier since its release in 2016. We provide new evidence of overfitting in the original AVClass evaluation results due to the use of poor reference data. We also demonstrate the ability to compare modified versions of classifiers using an AGTR by making minor modifications to AVClass and assessing their benefits or drawbacks. Evaluating such nuanced modifications was not previously tenable due to the lack of large reference datasets. The ability to compare the impact of model adjustments immediately that are otherwise hard to detect is of significant value in this domain, as production changes usually require months to obtain customer feedback or through "phantom" deployments (i.e., a new model is deployed alongside a previous model, but the new results are recorded for evaluation and comparison).

### 4.1 Testing AVClass Results Using an AGTR

Sebastian *et al.* [9] evaluated AVClass using five malware reference datasets. Because security vendors frequently refer to malware families by different names, the family names used by AVClass do not match those used by the reference datasets. Therefore, although AVClass is a classifier, Sebastian *et al.* could not compute its accuracy and chose to use precision and recall instead.

Precision and recall scores for the default version of AVClass are shown in Table 1. The row entitled MalGenome* is a modified version of the MalGenome dataset where labels for six variants of the DroidKungFu family are corrected. We call attention to the high variation in evaluation results - the precision of AVClass ranges from 0.879 to 0.954 and its recall ranges from 0.680 to 0.983. It is clear that due to these inconsistencies the evaluation results for AVClass are already suspect. To confirm this, we test the evaluation results of AVClass using the method described in subsection B.1.

Table 1: AVClass Evaluation

| Dataset | Precision | Recall |
|---|---|---|
| Drebin | 0.954 | 0.884 |
| Malicia | 0.949 | 0.680 |
| Malsign | 0.904 | 0.907 |
| MalGenome* | 0.879 | 0.933 |
| Malheur | 0.904 | 0.983 |

To construct an AGTR we use a portion of the VirusShare dataset [10]. The full VirusShare corpus contains 38,700,816 unlabeled malware samples dated between June 2012 and the time of writing. The VirusShare dataset is broken into chunks, and new chunks are added to the dataset regularly. We were provided with antivirus scan reports for chunks 0-7, which consists of 1,048,567 malware samples [11]. These scans were collected between December 2015 and May 2016 by querying the VirusTotal API [12]. We ran AVClass under default settings to obtain predicted family labels from each scan report. We produced a predicted clustering $C$ by assigning all malware samples with the same AVClass label to the same cluster. Malware samples for which no label could be determined were assigned to singleton clusters. Next, we created a peHash AGTR $\hat{R}$ from VirusShare chunks 0-7. Following our recommendation in subsection 3.1, we choose $\hat{e} = 10,000$ for the AGTR, which allows for an error rate of up to approximately one percent during the AGTR construction process.

Using this peHash AGTR we obtained the results that *Precision(C, $\hat{R}$)* $- \frac{\hat{\epsilon}}{m} = 0.229$ and *Recall(C, $\hat{R}$)* $+ \frac{\hat{\epsilon}}{m} = 0.895$. As a result of our analysis, we find that AVClass has an accuracy no greater than 0.895. The precision lower bound of 0.229 seems to be very loose considering that the smallest precision in Table 1 is 0.879. We attribute this to the moderate false negative rate of peHash; an AGTR construction technique that is better able to group data points should yield a tighter bound. The similarity between the recall upper bound and the reported recall results shows that although our peHash AGTR could be improved, the bounds are non-trivial. Designing improved methods for constructing AGTRs from malware datasets is an issue for future work. The Malsign, MalGenome, and Malheur datasets in Table 1 all have recall values exceeding the upper bound found using the peHash AGTR; the values for MalGenome and Malheur significantly so. Because VirusShare chunks 0-7, containing over a million malware samples from thousands of families, is significantly larger and more diverse than the Malsign, MalGenome, and Malheur datasets, we believe that evaluation results produced using those datasets are overfit to the labeling difficulties we discussed in Appendix C.

### 4.2 Comparing Modified Versions of AVClass

In this section we show that a peHash AGTR can be used to determine whether modifications to AVClass make a positive or negative impact on performance. In order to compare clustering algorithms or classifiers using an AGTR, they

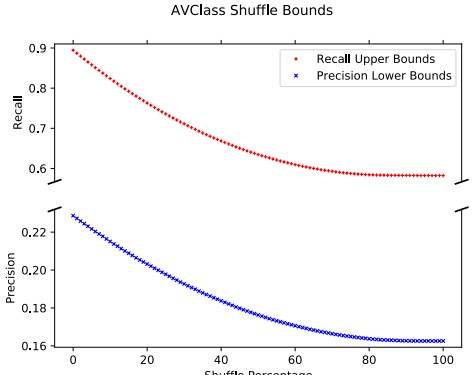

Figure 2: Precision and recall bounds of AVClass with respect to shuffle percentage. The x-axis of each figure shows the percentage of data points whose cluster membership has been shuffled. The y-axis of each figure shows the value of the metric bound. As the shuffle percentage increases, our bounds adjust monotonically and at a near linear rate, slowing only after 80% corruption.

must meet the two conditions listed in subsection B.2. Because we are comparing AVClass to slightly modified versions of itself, the classifiers are similar enough that the first condition is met. For the second step, we must determine if changes in classifier performance are strongly correlated with the evaluation metric bounds. To perform this check we use the same predicted clustering $C$ and AGTR $\hat{R}$ from subsection 4.1. Next, we incrementally shuffle $C$ and compute the precision and recall bounds each time that an additional one percent of the data points have been shuffled. Figure 2 shows how the bounds change as the data points are shuffled. It is evident that both bounds worsen predictably and monotonically as the data points are shuffled. Precison and Recall bounds have a correlation of -0.956 and -0.940 respectively with the ratio of labels shuffled, each with a p-value $\leq 10^{-47}$. Since a higher shuffle percentage indicates a worse clustering, there is likely a strong connection between the bounds and the true metric values. Because these two conditions have been met, we conclude that it is valid to compare modified versions of AVClass using the AGTR evaluation framework. Next, we compare modified versions of AVClass to the original tool. The purpose of this exercise is to demonstrate that an AGTR can be used to quantify the relative benefits and trade-offs of each of these changes to AVClass in the absence of reference data.

### 4.2.1 Comparing Alias Resolution Methods in AVClass

It is common for different antivirus engines to refer to the same family of malware by different names. We call two names for the same malware family *aliases* of each other. One of the steps that AVClass performs while aggregating antivirus signatures is resolution of family aliases [9]. If aliases are not resolved properly, AVClass could produce erroneous labels. By default, AVClass uses a manually generated list of known aliases. AVClass also has a setting for generating a family alias map based on families that have a high co-occurrence percentage within a corpus of antivirus scan results. Generation of the family alias map is controlled by the parameters $n_{alias}$, which is the minimum number of malware samples two tokens must appear in together, and $T_{alias}$, which is the minimum co-occurrence percentage.

To investigate how alias replacement affects label quality, we provide AVClass with three different family alias maps and use it to label VirusShare chunks 0-7. The first map is the one packaged within AVClass that is used by default. We generate the second map using the recommended parameter values $n_{alias} = 20$ and $T_{alias} = 0.94$, which were chosen empirically by Sebastian *et al* [9]. The third map was generated using the stricter parameter values $n_{alias} = 100$ and $T_{alias} = 0.98$ listed in the AVClass documentation.

Table 2 shows the precision lower bound ("Precision LB") and recall upper bound ("Recall UB") for AVClass using the three family alias mappings. Generating a map using the recommended parameters yields both higher precision and recall bounds than the default one. Gener-

Table 2: Alias Resolution Bounds

| Alias Preparation | Manual | Recommended | Strict |
|---|---|---|---|
| Precision LB | 0.229 | 0.230 | 0.233 |
| Recall UB | 0.895 | 0.897 | 0.894 |

ating a family alias map using the stricter parameters results in the highest precision bound but the lowest recall bound. All bounds are very similar, so none of the family alias maps appear to be significantly better than the others.

### 4.2.2 Adding a Threshold to AVClass' Plurality Voting

For the next modification, we add a plurality threshold to AVClass. By default, AVClass determines the label of a malware sample by selecting the plurality family proposed by the antivirus engines in a scan report [9]. Rather than using simple plurality voting to determine the label, we modify AVClass to require that the number of votes for the plurality family exceeds the number of votes for any other family by a given threshold. For example, if the plurality threshold is two, the plurality family must recieve at least two more votes than any other family. If no family meets this condition, AVClass outputs no label for that sample. We use the modified version of AVClass to label VirusShare chunks 0-7 with plurality thresholds between 0 and 5.

Table 3: Plurality Threshold Bounds

| Threshold | Precision LB | Recall UB |
|---|---|---|
| 0 | 0.229 | 0.895 |
| 1 | 0.276 | 0.881 |
| 2 | 0.332 | 0.860 |
| 3 | 0.442 | 0.829 |
| 4 | 0.511 | 0.803 |
| 5 | 0.565 | 0.780 |

Table 3 displays the precision and recall bounds of AVClass using different plurality thresholds. Note that a plurality threshold of zero is equivalent to the default version of AVClass. As the plurality threshold is raised, the precision lower bound significantly increases, indicating that higher thresholds reduce the number of false positives. However, raising the plurality threshold creates a trade-off, as it causes the recall (and hence accuracy) upper bounds to decrease to a lesser degree. This is largely due to the growing number of unlabeled malware samples contributing to the false negative rate. Thresholds above three may be useful for classifiers that require a very high precision. A threshold of one or two may offer a higher precision than the default version of AVClass without sacrificing a significant amount of recall. Since different applications of malware classification may require either a low false positive rate or a low false negative rate our findings indicate how designers of malware classifiers can adopt a suitable voting strategy.

### 4.2.3 Removing Heuristic Antivirus Signatures in AVClass Voting

When normalizing an antivirus signature, AVClass treats each token within the signature independently. However, we believe that incorporating contextual information from each token could improve AVClass' labeling decisions. A simple example of this is using context from tokens that indicate that the antivirus signature is a "heuristic". We believe that heuristic signatures are more likely to include inaccurate family information. To test this, we have identified eight tokens that indicate that an antivirus signature is a heuristic. We modify AVClass to exclude any AV's result if it contained any token in the set $\{gen, heur, eldorado, behaveslike, generic, heuristic, variant, lookslike\}$.

Table 4 shows the evaluation metric bounds for the default version of AVClass ("Default") and the modified version of AVClass where heuristic (abbreviated "Heur") antivirus signatures are not counted towards the plurality vote ("Heur Removal"). Simply ignoring common heuristic antivirus signatures substantially raises the precision

Table 4: Heuristic Removal Bounds

|  | Default | Heur Removal |
| --- | --- | --- |
| Precision LB | 0.229 | 0.250 |
| Recall UB | 0.895 | 0.889 |

bound of AVClass from 0.229 to 0.250. This comes at the cost of a minor increase in false negatives, as indicated by the slight drop in the recall bound. This confirms our suspicions that heuristic antivirus signatures often contain inaccurate family information. A more sophisticated method for handling heuristic signatures could offer even further improvements to AVClass.

## 5 Discussion and Conclusion

We now discuss limitations of this work and areas of future research. The foremost limitation of the AGTR evaluation framework is that constructing an AGTR requires domain knowledge of a problem space; there is no general strategy for constructing one. Evaluating the error rate of an AGTR construction technique is another challenging and open-ended problem. Finally, comparing models using an AGTR can only be done in limited cases.

We believe that this work has a multitude of avenues for future research. There are certainly many fields that could benefit from the AGTR evaluation framework, especially those where obtaining reference labels is difficult or time-consuming but grouping similar data points can be done easily. Our method currently relies on existing tools, which is valuable due to the epistemic uncertainty involved in estimating error rate $\hat{\epsilon}$. Developing new AGTRs that explicitly inform the value or range of $\hat{\epsilon}$ is of interest. Similarly, developing the remaining bounds on precision and recall, and on other metrics, are open problems.

We have established a method for computing bounds on precision, recall, and accuracy without a reference dataset, which becomes all the more important as datasets become too large to manually validate. In addition, we have designed a litmus test that uses AGTRs to identify biased evaluation results produced by low-quality reference datasets. We show that AGTRs can be used to evaluate the impact of changes made to a model. We identify malware family classification as a field where the AGTR validation framework provides value, provide an implementation for constructing an AGTR using peHash, and apply the AGTR evaluation framework to the AVClass malware labeler. There is no shortage of other problem spaces with inadequate reference datasets, and we believe that this work can be used to improve the evaluation process for them as well.

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
