# OpenReview forum: "A Framework for Cluster and Classifier Evaluation in the Absence of Reference Labels"
_NeurIPS.cc/2021/Track/Datasets_and_Benchmarks/Round1 — Submitted to NeurIPS 2021 Datasets and Benchmarks Track (Round 1)_

### Official Review · Reviewer_ACdN · 2021-07-03
**Interesting idea but more works need to be done.**

**Rating:** 5
**Confidence:** 3
**Clarity:** The paper writes well and it is easy …

**Strengths:**

1. The idea of firstly creating an incomplete sub-graph of ground-truth reference labels (AGTR) and then estimate the precision and recall bounds is interesting.
2. The proposed evaluation can be applied to a variety of learning settings (eg, clustering and multi-class classification)

**Weaknesses:**

1. The theorems 1 - 4 seem to be pretty obvious. I would suggest replacing them with propositions.

2. The bounds in theorems 5 and 6 can be loose when $\epsilon$ is large. Moreover, there is no guarantee on why $\hat{\epsilon} > {\epsilon}$.

3. As the author mentioned in the paper, I share the same concern about no general strategy for constructing an AGTR. Given the same dataset, different constructing methods will result in different estimation bounds based on the proposed evaluation methods.

4. Since domain knowledge is needed to construct a good AGTR. Why not considering subsample the reference dataset and then annotate them for evaluation?

5. Lastly, this paper only evaluates their approach on one user-case study. It would be more convincing to have more real-world case studies to validate the effectiveness of their proposed approach.

**Additional Feedback:**

This paper studies a very interesting problem. However, I feel the current version still needs some improvement.  I'd like to see a general way to construct AGTR.

For the experimental part, I would suggest the authors start from some synthetic data sets to validate the tightness of estimation bounds. Perhaps the synthetic dataset can be generated based on some hierarchical clustering algorithms.

Lastly, I think the paper reads well and most parts of the paper are easy to follow (I like figure 1). One minor issue I noticed is that many appendix contents are referenced in the main paper. This may reduce the readability of the main paper.

**Correctness:**

The definition of AGTR is sound. But the way to construct AGTR still needs some improvement. The experimental evaluation part should include more user-case studies to validate the effectiveness.

**Documentation:**

Yes, the authors have provided sufficient information to support reproducibility.

**Ethics:**

No ethics issues.

**Relation To Prior Work:**

The paper lacks a discussion about related work or settings.

**Summary And Contributions:**

This paper proposes a new evaluation framework when the labels in the reference datasets are of low quality. Specifically, they introduce a strategy called approximate ground truth refinement (AGTR) to estimate the error bounds on the precision and recall of clustering or classification algorithms. With some domain knowledge in the construction of an AGTR, they use one case study in malware classification to illustrate the effectiveness of the proposed measure.

---

### Official Review · Reviewer_2URT · 2021-07-03
**A Malware Detection Dataset**

**Rating:** 5
**Confidence:** 2

**Strengths:**

-The paper derives seemingly useful bounds on the performance of systems in practical problem settings, namely in incomplete/partially labelled test datasets.
-The proposed bounds are shown to be highly correlated with the ground truth with label shuffling experiments.
-The application domain of cyber security/malware is very important in practice.

**Weaknesses:**

Due to the highly specialized nature of the dataset I am not able to judge the significance/novelty of the proposed framework. I believe this work is better suited for a more specialized venue (e.g. cyber security conferences). The authors explicitly point out that domain knowledge is required in order to apply their framework.

**Additional Feedback:**

I recommend submitting this work to more specialized conference, or an applications centered conference such as KDD.

**Clarity:**

Although the paper is well written, I believe it is better suited for a more specialized audience. I am not familiar with malware detection features/datasets/standards. I would suggest adding some intuitive explanations to complicated statements such as Property 1 (Line 112).

**Correctness:**

The derivations seem elementary, although I did not go through the appendix in detail.
The empirical evaluation suggests that they are correct.

**Documentation:**

Yes

**Relation To Prior Work:**

The authors did not include an explicit prior works section, and because I am not familiar with this area, it is not clear what the relationship is other works in this field.

**Summary And Contributions:**

The paper proposes a framework (AGTR) for estimating/bounding model performance with small/incomplete reference datasets.
The proposed method applies to both clustering and classification algorithms. The proposed framework derives bounds on precision/recall performance on datasets with missing/incomplete labels, without needing to infer the ground truth label for the reference dataset. The proposed framework is applied to analyze the performance of a popular malware labeling tool - AVClass.
The proposed framework is empirically validated with label shuffling experiments to show that the derived bounds on precision/recall are highly correlated with the ground truth.

---

### Official Review · Reviewer_t1wu · 2021-07-05
**Domain-specific conferences might be a better fit.**

**Rating:** 5
**Confidence:** 3
**Correctness:** The experiments look sound. I haven't…

**Strengths:**

- The AVClass experiment is extensively performed.

**Weaknesses:**

When the paper discusses the limitation of their approach, they mention that AGTR requires domain knowledge to construct. In my opinion, requiring domain knowledge to improve evaluation of clustering algorithms is inevitable. The limitation I find is that their experiments are only performed on the malware dataset. Since the framework is advertised in a way that is applicable to any domain, the experiment section should be more thorough in terms of the number of domains for the framework to be evaluated.

Because of this, it is hard to assess how this framework benefits the community other than the security domain. For example it's unclear how easy / difficult it is for practitioners to construct AGTR for other domains.

Nevertheless, the malware classification experiment is extensively discussed in this paper; with the current scope, more domain-specific conferences (i.e. security conferences) might be a better fit for this paper.


**Additional Feedback:**

NA

**Clarity:**

Minor: The explanation of AVClass tool (line1 page2) should be added when it's introduced.

**Documentation:**

Yes

**Relation To Prior Work:**

Related work section should be added to emphasize the paper's contribution and discuss the positioning of this paper in the context of prior work.

**Summary And Contributions:**

- The paper introduces a framework to evaluate clustering algorithms in the absence of reference labels.
- They provide a method to compute bounds on precision, recall, and accuracy using an Approximate Ground Truth Refinement (AGTR).
- In the experiment section, they demonstrate the effectiveness of the proposed framework using malware classification as an example.

---

### Decision · Program_Chairs · 2021-07-26

**Decision:**

Reject

**Comment:**

This paper provides a dataset for cluster and classifier evaluation in the absence of reference labels.  The reviewers found that the generality of the proposed dataset/benchmark beyond the one domain studied was not sufficiently validated.